# Overview of Cellular and Soluble Mediators in Systemic Inflammation Associated with Non-Alcoholic Fatty Liver Disease

**DOI:** 10.3390/ijms24032313

**Published:** 2023-01-24

**Authors:** Patrice Marques, Vera Francisco, Laura Martínez-Arenas, Ângela Carvalho-Gomes, Elena Domingo, Laura Piqueras, Marina Berenguer, Maria-Jesus Sanz

**Affiliations:** 1Institute of Health Research INCLIVA, University Clinic Hospital of Valencia, 46010 Valencia, Spain; 2Department of Pharmacology, Faculty of Medicine and Odontology, University of Valencia, 46010 Valencia, Spain; 3Endocrinology and Nutrition Service, University Clinic Hospital of Valencia, 46010 Valencia, Spain; 4Liver Transplantation and Hepatology Laboratory, Hepatology, HBP Surgery and Transplant Unit, Health Research Institute Hospital La Fe, La Fe University Hospital, 46026 Valencia, Spain; 5CIBEREHD, Spanish Biomedical Research Centre in Hepatic and Digestive Diseases, Carlos III Health Institute (ISCIII), 28029 Madrid, Spain; 6Department of Biotechnology, School of Agricultural Engineering and Environment, Universitat Politècnica de València, 46022 Valencia, Spain; 7CIBERDEM, Spanish Biomedical Research Centre in Diabetes and Associated Metabolic Disorders, Carlos III Health Institute (ISCIII), 28029 Madrid, Spain; 8Department of Medicine, Faculty of Medicine and Odontology, University of Valencia, 46010 Valencia, Spain

**Keywords:** NAFLD, MAFLD, NAFL, NASH, steatosis, fibrosis, inflammation, platelets, leukocytes, biomarkers

## Abstract

Non-alcoholic fatty liver disease (NAFLD) is currently the most prevalent chronic liver disease in Western countries, affecting approximately 25% of the adult population. This condition encompasses a spectrum of liver diseases characterized by abnormal accumulation of fat in liver tissue (non-alcoholic fatty liver, NAFL) that can progress to non-alcoholic steatohepatitis (NASH), characterized by the presence of liver inflammation and damage. The latter form often coexists with liver fibrosis which, in turn, may progress to a state of cirrhosis and, potentially, hepatocarcinoma, both irreversible processes that often lead to the patient’s death and/or the need for liver transplantation. Along with the high associated economic burden, the high mortality rate among NAFLD patients raises interest, not only in the search for novel therapeutic approaches, but also in early diagnosis and prevention to reduce the incidence of NAFLD-related complications. In this line, an exhaustive characterization of the immune status of patients with NAFLD is mandatory. Herein, we attempted to gather and compare the current and relevant scientific evidence on this matter, mainly on human reports. We addressed the current knowledge related to circulating cellular and soluble mediators, particularly platelets, different leukocyte subsets and relevant inflammatory soluble mediators.

## 1. Introduction

About one-third of the world’s population is currently overweight or obese [1] and, according to the last World Health Organization (WHO) obesity report, this percentage raises to 60% for European adults [2]. In fact, obesity is among nine global noncommunicable diseases that must be dealt with. Stratifying by risk, subjects with body mass index (BMI) cut-offs of 27.5 kg/m^2^ or higher are considered to have an increased risk of obesity-related conditions, including metabolic syndrome (MS), type 2 diabetes mellitus (T2DM) and non-alcoholic fatty liver disease (NAFLD) [3,4]. Indeed, it has been estimated that 70% of overweight individuals and 90% of obese individuals develop NAFLD. NAFLD is a mostly asymptomatic condition encompassing a spectrum of liver diseases. It is characterized by abnormal accumulation of fat in liver tissue (simple steatosis or non-alcoholic fatty liver, NAFL) that may progress to a more severe state, such as non-alcoholic steatohepatitis (NASH). Both are liver manifestations with a prevalence that has been increasing in Western countries [5]. Of note, the recent term metabolic dysfunction-associated fatty liver disease (MAFLD) has begun replacing NAFLD nomenclature. The replacement of the term non-alcoholic with metabolic dysfunction-associated is reflective of an effort to better define this heterogeneous pathology and avoid its possible trivialization or stigmatization [6,7]. Nevertheless, since the vast majority of the scientific literature included in this review was still based on NAFLD diagnostic criteria, the terms NAFLD, NAFL and NASH were used herein.

NAFLD, representing approximately 75% of all chronic liver diseases, is considered the most common chronic hepatic disease in Western countries, with a constant increase in prevalence and incidence [8]. According to a recent meta-analysis, the highest prevalence of NAFLD can be found in North and South America (35%), followed by Asia and Europe (30%) and then Africa (28%) [9]. Epidemiologic data from Spain demonstrate similar rates, with an NAFLD prevalence of 25.8% in the adult population [10]. The current increase in global NAFLD prevalence seems to be due to an expanded unhealthy and sedentary lifestyle based on reduced physical activity and/or long-term hypercaloric diet consumption among the world’s population [11].

On the other hand, NASH, a hepatic complication derived from the evolution of NAFLD, and with a worldwide prevalence close to 3–5%, is defined by the presence of steatosis and lobular inflammation with hepatocyte injury [11]. This more severe form is often accompanied by liver fibrosis which, in turn, may progress to a state of cirrhosis or hepatocarcinoma, both irreversible processes that often lead to the need for liver transplantation (LT) [3,12]. Preliminary data from the largest LT center in Spain show an increased prevalence of NASH-related cases, from 0.8% in 1997–2001 to 2.3% in 2012–2016 [13]. Despite its current high success rate (9 out of 10), LT remains a risky medical intervention, associated with several post-transplant complications that have intimately related to immune and metabolic status, e.g., renal failure, T2DM, cardiovascular disease, de novo tumors development and severe infections, many related to the chronic effects of immunosuppression [12]. According to a recent Spanish cohort study, survival at 1-year post-transplant was approximately 87.14%; this percentage decreased to 62.25% at 10-years post-transplant [14].

Data from the United Kingdom in 2018 showed that total health system costs due to diagnosed NASH were estimated to range between £200 to £359 million [15]. In Spain, the same year, the annual NASH-related national medical costs reached nearly €3.5 billion, corresponding to €2162 per patient annually. These numbers placed Spain as the country with the highest annual NASH-related national medical cost per patient when compared to other four European countries (United Kingdom, France, Italy and Germany) [16]. With NAFLD-related complications taken into account, it was estimated that medical costs could rise to more than €35 billion per year in Europe [17].

Along with the associated high economic burden, the high mortality rate among NAFLD patients raises interest, not only in novel therapeutic approaches, but also in early diagnosis and prevention, in order to reduce the incidence of NAFLD-related complications. NAFLD may be associated with a proinflammatory milieu, which is required to evolve into NASH and seems to be a trigger for the development of related complications [18,19]. Therefore, an exhaustive characterization of the immune and metabolic status of patients with NAFLD was essential, to better understand its complex pathophysiology and, in turn, to predict the disease prognosis and possible associated complications—in order to act at an earlier stage of the disease. Additionally, these findings could also aid the discovery of novel therapeutic targets to treat or prevent the progression of this metabolic disorder. In the present review, we attempted to gather and compare the current and relevant scientific evidence on this matter, focusing mainly on reports addressing human studies related to both circulating cellular and soluble mediators.

## 2. Role of Systemic Inflammation on NAFLD Pathogenesis and Progression

Although evidence points to a strong association of NAFLD with systemic inflammation, mandatory in the progression to NASH, the exact immunological mechanisms that cause inflammation in this complex metabolic disease are poorly defined [20]. Over time, some hypotheses have been proposed. The first was described in 1998, by Day and James, as the two-hit hypothesis, postulating that the so-called first hit was responsible for steatosis development, whereas the second hit resulted in the progression into steatohepatitis [21]. Triglycerides (TGs) stored in the liver usually stem from the diet, de novo lipogenesis (DNL) or lipolysis of white adipose tissue. When an excessive liver accumulation of TGs or free fatty acids (FFA) from visceral adipose tissue occurs, hepatic insulin resistance is induced and steatosis is established (first hit). Then, free radicals are generated (e.g., reactive oxygen species (ROS)), inducing lipid peroxidation and oxidative stress and leading to mitochondrial dysfunction, hepatocyte damage, inflammation and steatohepatitis development [21,22]. However, this hypothesis has been considered too simplistic among the scientific community, given that other factors also seem to play a pivotal role in this complex pathophysiology.

Thus, in 2010, Tilg and Moschen described the hypothesis of multiparallel hits [23]. According to this hypothesis, liver inflammation, and consequently the development of NASH, is promoted by several hits derived from gut and adipose tissue (AT), involving endoplasmic reticulum stress, proinflammatory mediators’ release (e.g., adipokines and other cytokines) and innate immunity [22,23]. The release of proinflammatory cytokines, such as tumor necrosis factor-α (TNFα), from AT, is increased in obese patients or patients with insulin resistance [23], which seems to play a crucial role in liver steatosis, inflammation and fibrosis [24,25]. Moreover, classical innate immune cells (such as Kupffer cells, dendritic cells, neutrophils and innate lymphoid cells) are not the only cells involved in NASH pathogenesis; some non-innate immune cells, including hepatocytes and liver sinusoidal endothelial cells, may play immune cell-like functions when stressed [26]. A decade later, in 2021, Tilg et al. highlighted the crucial role of AT contribution as well as gut microbiome, dietary components and genetic pathways in NASH development and progression [27]. According to this multiparallel hits hypothesis update, obesity-induced dysfunction of the AT-liver axis (in energy homeostasis) seems to be the trigger for liver inflammation and injury.

Indeed, beyond its energy storage functions, AT is now considered an important endocrine organ, given that it produces and releases cytokine-like hormones, known as adipokines. This family of low molecular weight bioactive peptides has proven pleiotropic functions as hormones and cytokines with both pro- and anti-inflammatory activities. The difference between the detrimental or beneficial actions of systemic inflammation in the context of NASH is determined by the different expression of individual inflammatory mediators, the type of inflammatory cells present and, even more importantly, by the stage of the disease [28]. As mentioned above, NAFLD is related to obesity, with more than 80% of NAFLD patients in Europe and North America being obese [29]. Indeed, although 5–8% of NAFLD patients are lean, they still have abnormal glucose tolerance and excess of visceral adipose tissue [30]. During obesity, adipocytes undergo hypertrophy, which may cause them to stress and/or rupture, releasing proinflammatory adipokines. This, in turn, leads to a local inflammatory phenotype characterized by the recruitment and activation of immune cells [31]. Namely, the increased AT adipokine and chemokine expression (e.g., leptin, visfatin, resistin, MCP-1/CCL2, RANTES/CCL5, CCL13, TNFα, IL-1β and IL-6) promotes increased immune cell infiltration (mainly macrophages, CD4+ and CD8+ T cells, dendritic cells and natural killer cells) that contributes, not only to local, but also systemic, inflammation [23]. This proinflammatory environment also exhibits regulatory functions in T lymphocyte differentiation, inducing differentiation toward proinflammatory subclasses (e.g., Th1 and Th17) [32].

In addition to AT, other factors of extrahepatic origin, such as gut microbiota, may cause liver inflammation in NAFLD. Many studies have pointed out the crucial role that the gut-liver axis seems to be gaining in the complex NASH pathogenesis. Altered intestinal permeability may lead to increased bacteria translocation (some of them abnormally abundant in the gut of obese patients) and bacteria-derived products, initiating or promoting liver inflammation [33,34]. Several bacterial components, such as lipopolysaccharide (LPS) and other pathogen-associated molecular patterns (PAMPs), promote inflammation. For example, LPS activates toll-like receptor 4 (TLR4) of immune and parenchymal cells of the liver, which, in turn, triggers an inflammatory cascade that may involve activation of nuclear factor (NF)-κB, activator protein 1 (AP-1) and interferon regulatory factor 3 (IRF3) [35]. LPS-activated Kupffer cells, via the Myeloid differentiation primary response 88 (MyD88)-dependent signaling pathway, produce proinflammatory cytokines (IL-18, IL-1β, and IL-12) that amplify the cytotoxic activity of NK and CD8+ T cells, and hepatic TNFα and ROS formation, together with insulin resistance. In addition, Kupffer cells increase the production of transforming growth factor-β (TGF-β) which, through the interaction with its receptor in hepatic stellate cells (HSCs), enhances fibrogenesis. Finally, other TLRs have also been associated with the development of NASH, including TLR2, TLR5 and TLR9 [35].

Overall, a dysfunctionality of both AT-liver and gut-liver axes drives, not only an abnormal lipid accumulation in the liver (steatosis), but also liver inflammation, a trigger of NASH development. In the present review, the contribution of adipokines to NASH-associated systemic inflammation was not addressed, given that this subject was covered in a recent review from our group [36]. Hereafter, the contributions of platelets, leukocyte subsets and several soluble mediators in this complex metabolic disorder were addressed. Contributions of the main cellular and soluble mediators in NAFLD development are illustrated in Figure 1.

## 3. Contribution of Platelets in NAFLD

Beyond the well-known role of platelets in the regulation of hemostasis under both physiological and pathological conditions, recent studies have described additional roles of platelets that seemed to be independent of their main functions, through interactions with other immune cells (reviewed in [37]).

Platelets are closely related to the liver, since they are mainly produced by this organ during fetal life. From birth, bone marrow is the main source of platelets, where they develop from their progenitors, the megakaryocytes. Both megakaryocyte maturation and platelet production are tightly regulated by the action of thrombopoietin (TPO), a glycoprotein hormone produced by the liver and the kidney [37].

The relevance of liver-derived TPO (constitutively synthesized by both the liver parenchymal and the sinusoidal endothelial cells) relies on the strong association of severe liver diseases with thrombocytopenia, defined as a platelet count less than 150 × 10^9^/L. It is noteworthy to mention that hypersplenism secondary to portal hypertension also contributes to thrombocytopenia through splenic platelet sequestration in advanced liver disease [38]. Thus, the prevalence and severity of thrombocytopenia seem to correlate with the stage of liver disease. Indeed, patients with liver diseases present an increased risk of bleeding and require platelet transfusions. Thus, in an NAFLD context, the associated chronic inflammatory state leads to liver parenchymal damage and fibrosis, resulting in a reduction in TPO production and the consequent decrease in circulating platelet count [39].

Nevertheless, a recent study pointed out a hepatic accumulation of platelets in patients with NASH, and similar results were obtained in a NASH-mice model (6-month C57BL/6 mice fed with a choline-deficient high-fat diet) [40]. Of note, no differences were found regarding the circulating platelet count between the NASH-murine model and the control-diet subjected group. Interestingly, this hepatic accumulation of platelets was not observed in a steatosis-murine model (6-month C57BL/6 mice fed with a 45% kcal high-fat diet, HFD), suggesting that platelets play a hepatic pivotal role in steatosis progression to NASH [40]. In agreement with these results, an antiplatelet treatment (aspirin-clopidogrel) attenuated steatosis (e.g., reduction of hepatic TG content, serum cholesterol levels and NAFLD Activity Score, NAS), reduced liver damage (decreased alanine aminotransferase (ALT) levels) and abrogated hepatic immune cell infiltration (T lymphocytes and macrophages) in the NASH-murine model [40]. Malehmir et al. also highlighted the relevance of platelet glycoprotein Ibα (GPIbα), required for hepatic TPO production [41], as a key player for NASH induction. In fact, the neutralization of this glycoprotein, or its genetic dysfunction, reduced steatosis, fibrosis and liver damage [40]. Moreover, Arelaki et al. also described the presence of larger platelet aggregates in liver biopsies of patients with NASH than in control specimens, which confirmed NASH-associated thromboinflammation and could explain the reduction of circulating platelet counts reported in these patients [42].

While the reduced peripheral blood platelet counts in NASH seem to be clear, it remains controversial in NAFLD. In this regard, some studies described reduced circulating platelet counts in NAFLD patients compared to control cases [43,44], and increased risk of platelet count reduction compared to non-NAFLD population [45]. Garjani et al. suggested that it could be useful as a biomarker to classify NAFLD severity, but not as a sole test [46]. In contrast, Saremi et al. found no differences between NAFLD patients and healthy controls in this parameter [47]. In fact, the prevalence of thrombocytopenia in NAFLD varied according to the method used for NAFLD diagnosis and the nature of the study. Nowadays, NAFLD can only be unequivocally diagnosed through a biopsy, which was only used in one study [44]. In the others [43,45,46,47], noninvasive approaches were employed.

On the other hand, another widely-studied platelet parameter is the mean platelet volume (MPV), which is considered an indicator of platelet size and function. Large platelets contain a higher density of prothrombotic material, leading to the release of substances that amplify platelet activation, adhesion and proliferation, such as adenosine diphosphate (ADP) or thromboxane A2 (TxA2). These, in turn, contribute to their greater aggregability and worse response to antiplatelet therapy [37]. Notably, while several studies indicated that MPV seemed to be significantly increased in the population with NAFLD with or without obesity [43,45,47,48], others did not [44]. Interestingly, MPV values showed a correlation with the severity of inflammation and the degree of fibrosis (Table 1) [48,49]. It seemed that different factors were responsible for the MPV increase in NASH subjects with liver fibrosis, such as insulin resistance and AT-related proinflammatory cytokines (including IL-1β, IL-6 and TNFα) [50]. It was also shown that MPV, considered a risk factor for atherothrombosis, was a prognostic biomarker for cardiovascular diseases [51]. In this line, platelet activation is intimately related to MPV and has also been associated with increased leukocyte-endothelium interactions—a key event for atherosclerosis development, at least in some metabolic diseases [52,53,54].

Different platelet-related parameters have also been investigated in the context of NAFLD, such as platelet distribution width (PDW) and plateletcrit (PCT). While PDW characterizes the range of size difference among platelets, PCT is the volume, in percentage, occupied by platelets in blood. A wide PDW may be a sign of inflammation and platelet activation, and seems to convey more information than MPV regarding platelet reactivity [44]. However, similarly to peripheral platelet counts, PDW variation in NAFLD remains controversial. While some studies showed that PDW was significantly higher in patients with NAFLD than in control groups [43,48], other studies detected no differences between these populations [44,47]. Regarding PCT, it was reported to be associated with advanced fibrosis in chronic viral hepatitis, and a potential prognostic marker in the early detection of NAFLD [55]. Nonetheless, while some studies reported higher levels of PCT in NAFLD patients than in controls [43,55], Oral et al. found the opposite [44]. These discrepancies may rely on the inclusion/exclusion criteria used for the selection of the control group.

Taking all these data together, MPV, PDW and PCT should not be considered relevant peripheral biomarkers of NAFLD development and progression. However, no doubt remains about the relevance of platelets in this disease, since both thrombocytopenia and increased platelet infiltration in the liver are common features in NAFLD [39,42,43,44,45]. Therefore, more exhaustive studies are required to better understand the role of platelets in this complex pathology.

## 4. Contribution of Leukocytes in NAFLD

### 4.1. Neutrophils

Neutrophils are the most abundant leukocytes in human blood and the first line of immune defense against infection or injury, contributing to the acute inflammatory response. This leukocyte subset is continuously released from the bone marrow, and their release into the peripheral blood is tightly regulated by different molecules, including granulocyte colony-stimulating factor (G-CSF), and ligands of CXC chemokine receptors (CXCR), such as CXCR2 and CXCR4 [56]. They are considered the main players in the innate immune response; however, despite their extensive studied contribution to acute liver injury, little is known about their role in NAFLD [57].

In this regard, several in vivo experimental studies were carried out. Intraperitoneal administration of a neutrophil-neutralizing antibody (anti-Ly6G) in a NASH-murine model (10-week HFD-fed C57BL/6 mice) reduced neutrophil liver infiltration, ameliorated metabolic features (reduction of fasting glycemia, hepatic TG content and transaminase activity), decreased hepatic inflammation (abrogation of macrophage infiltration and expression of proinflammatory cytokines such as TNFα, IL-6 and MCP-1) and profibrotic environment (reduction of profibrotic cytokine hepatic expression, such as TGF-β and α-smooth muscle actin (α-SMA)) [58]. Interestingly, the role of neutrophils in this context seemed to partially rely on neutrophil elastase, given that its deficiency (using a knockout murine model) improved the lipid profile and reduced hepatic damage (reduced transaminase activity, steatosis and NAS) as well as liver inflammation (decreased macrophage infiltration, TNFα and MCP-1 expression) [59]. Unlike the results found by Ou et al. [58], interleukin (IL)-6 expression was not significantly affected [59]. These discrepancies could account for the differential NASH induction or the involvement of other neutrophil-related mediators beyond elastase, such as myeloperoxidase (MPO). Indeed, both TNFα and IL-6 expression were downregulated in an MPO-deficient NASH-murine model, and this anti-inflammatory response was accompanied by a decrease in both neutrophil and lymphocyte hepatic infiltration, as well as by an improvement in NASH-related features, such as liver cholesterol content and the degree of fibrosis [60]. Accordingly, both MPO-deficiency and MPO pharmacological inhibition reduced liver damage, steatosis and fibrosis, as well as plasma levels of ALT, in a murine model of NASH, which could be explained by the existence of an MPO-dependent pathway for inflammation and apoptosis in this metabolic disease [61]. Moreover, there is additional evidence of MPO contribution to fibrosis development. MPO appeared to activate HSCs and to upregulate the profibrotic mediators TGF-β and α-SMA, crucial events for collagen production (Figure 1) [62]. In humans, plasma levels of MPO were found to be increased in NASH subjects, compared to healthy volunteers [61,63] or to patients with simple steatosis [64]. Furthermore, MPO mRNA hepatic expression was positively correlated with risk factors for NASH progression, such as body mass index (BMI) and the percentage of glycated hemoglobin (Table 1) [61].

On the other hand, neutrophil extracellular traps (NETs), a matrix involved in pathogen capture and destruction, appear to play a role in the early stages of NAFLD, even before monocyte-derived macrophage infiltration [65]. In particular, the dissolution of NETs resulted in mice protection from liver inflammation (reduction of TNFα and IL-6 expression and macrophage infiltration) and damage (diminished ALT levels), which ultimately contributed to NAS reduction [65].

Regarding some cell adhesion molecules (CAMs) expressed on the neutrophil surface, L-selectin (also known as CD62L, which is involved in the initial rolling) was found to be overexpressed on neutrophils from NASH patients, compared to those found in healthy controls or patients with simple steatosis. Despite these findings, no differences were observed in the neutrophilic expression of CD11b, an integrin involved in leukocyte adhesion and transmigration through the endothelium [63].

Furthermore, it is known that metabolic abnormalities lead to immune imbalances in peripheral blood and liver. They are manifested at the cellular level by an increased ratio of T helper (Th)17 lymphocytes to T regulatory (Treg) cells (ratio Th17/Treg) and by the dominance of neutrophils over lymphocytes. Therefore, one of the most studied parameters in liver diseases is the neutrophil/lymphocyte ratio (NLR) [66]. According to most of the current literature, elevated NLR values have been associated with greater severity of the disease, the evolution of liver fibrosis and the prediction of mortality in NAFLD [48,57,63,66,67]. However, Kara et al. found no association between NLR and the severity of liver inflammation or fibrosis in patients with NAFLD [68]. Nevertheless, NLR values were found to be higher in NASH patients than in healthy controls, or even in subjects affected by hepatitis B or C [56,67]. Additionally, it seems that NLR is a better predictive marker than C-reactive protein (CRP) for active chronic liver disease, and can be considered as an independent variable for predicting the occurrence of necroinflammation and fibrosis in NASH [67].

Along with imbalances in NLR, an altered Th17/Treg ratio also contributes to the upregulation of the above-mentioned proinflammatory (e.g., IL-6, TNFα) and profibrotic (e.g., TGF-β) cytokines, which in turn leads to a hyperactivation of the IL-17 axis, implicated in the progression of NAFL to NASH [66]. Indeed, it is known that hepatic human neutrophils, especially in patients with advanced NAFLD, are a relevant source of IL-17 [57,69]. IL-17 plays an important role in granulopoiesis and participates in the recruitment and infiltration of neutrophils in the initial organ injury. IL-17 also induces the production of neutrophilic cytokines and chemokines, amplifying the neutrophilic response, which aggravates the lesion (Figure 1) [66]. Accordingly, neutrophil depletion reduced liver inflammation and further complications in the context of NASH in mice [58]; however, these results could have been biased, given that these cells constitute the front line of host defense against infections.

Altogether, neutrophils appear to play a relevant role in both NAFL and NASH progression. In these diseases, neutrophil activation, increased CAMs expression on their surface, and generation and release of neutrophil-related cytokines, chemokines, enzymes and other intracellular products were detected.

### 4.2. Monocytes

Monocytes are highly plastic leukocytes that play a crucial role in host defense and tissue homeostasis, especially in bacterial and fungal elimination through phagocytic, oxidative and cytokine-producing responses [70]. This leukocyte subset can be divided into three functionally distinct phenotypes based on the differential expression of the surface markers CD14, CD16 and CCR2 [52,70,71]:classical monocytes (CD14++CD16−CCR2+, also known as Mon1 subtype monocytes), represent approximately 85% of monocytes in peripheral blood. They possess a high phagocytic capacity and proinflammatory properties;intermediate monocytes (CD14++CD16+CCR2+, also known as Mon2 subtype monocytes), which constitute around 5% of the total monocytes in peripheral blood;nonclassical monocytes (CD14+CD16+CCR2−, also known as Mon3 subtype monocytes), represent 10% of the total monocytes in peripheral blood.

Different studies demonstrated that the total circulating leukocyte counts were elevated in patients with NAFLD, as compared to control subjects, partly due to an increase in the monocyte fraction [72,73]. Interestingly, the percentage of the Mon1 subset was found to be lower in NAFLD patients and, while Zhang et al. described an increase in the Mon2 fraction, Wang et al. documented a higher Mon3 fraction in patients with NAFLD [73,74]. Compiled evidence suggested a link between adiposity, inflammation and intermediate/nonclassical monocytes (CD16+). However, it remains unclear whether the unbalance in monocyte subtypes is a consequence of—or a contributor to—this inflammatory response. Nevertheless, monocyte fraction (along with BMI, waist circumference and plasma levels of TNFα) turned out to be an independent risk factor for NAFLD, and could become a potential prognostic biomarker in this context [73,74].

In agreement with these observations, CCR2 expression in monocytes was significantly lower in NAFLD patients, compared to control subjects [70], which was perhaps the consequence of imbalances between Mon1 and Mon3 fractions, as previously described [73]. In this regard, the development of novel drugs has shown promise. For instance, Cenicriviroc, a dual antagonist of CCR2 and CCR5 receptors, yielded satisfactory results in the first year of its phase II clinical trial, showing a significant reduction in systemic inflammation and markers of inflammation in NASH [75]. TLR6 was also found to be overexpressed in the monocytes of patients with NAFLD, compared to obese subjects with normal liver biopsies [76]. TLR6 is associated with PAMPs recognition, crucial for innate immunity activation against infectious agents. Along this line, since liver blood flow comes directly from the intestinal portal circulation, intimately linking the gut and liver, the gut microbiota profile could influence liver histology through TLR6 activation. Indeed, TLR6 deregulation found in NAFL and NASH patients seems to contribute to the worsening of liver inflammation, and has been pointed out as a potential peripheral biomarker of NASH severity [76].

Regarding cardiovascular complications, different studies have shown a higher incidence of subclinical atherosclerosis in patients with NAFL/NASH compared to controls (reviewed in [19]). Monocytes play a central role in atherosclerosis [52] and could play a relevant role in the development of cardiovascular diseases in NAFLD [72,73]. A possible molecular player connecting these pathologies is L-selectin/CD62L, involved in leukocyte-endothelium interactions. Its expression has been found to be increased in monocytes of patients with decompensated cirrhosis [70].

In summary, high monocyte counts due to a greater intermediate/nonclassical monocyte fraction appear to contribute to the development of NAFLD, its progression toward NASH and/or the development of cardiovascular complications.

### 4.3. Lymphocytes

Lymphocytes are certainly the most relevant leukocytes of the adaptive immune system. According to their activity, they are classified as T-lymphocytes (CD3+ cells), responsible for the cell-mediated responses, or B-lymphocytes (CD19+ cells), which participate in humoral/antibody responses. In turn, according to their physiological functions, T-lymphocytes are divided into different subtypes: cytotoxic T-lymphocytes (Tc or CD8+ cells) and different subsets of T-helper cells (Th or CD4+ cells), including Th1, Th2, Th17 and T regulatory lymphocytes (Treg), which seem to be involved in the pathogenesis of NAFLD [32]. T-helper cells are the main regulators of immune processes, orchestrating the effector functions of B-lymphocytes, cytotoxic T-lymphocytes and phagocytes. In addition, other different T-lymphocyte subtypes are associated with the innate immune system, including natural killer T-lymphocytes (NKT cells), γδ T-lymphocytes and mucosal-associated invariant T cells (MAIT cells) [20,32].

In recent years, the complex bidirectional interaction between T-lymphocytes and neutrophils has become evident, with neutrophils playing an important role in the modulation of T-lymphocyte immunological response. Antonucci et al. observed that neutrophils from NASH patients were able (ex vivo) to suppress the proliferation and activation of autologous CD4+ and CD8+ T-lymphocytes more than neutrophils from healthy donors or those with NAFL [63]. Potent suppression of CD4+ and CD8+ T-lymphocyte proliferation and activation could, over time, induce inadequate immune surveillance of hepatic damage, making patients more susceptible to NAFLD progression.

Positive correlations between lymphocyte aggregates (rich in T-lymphocytes), in number, size, lobular inflammation score or fibrosis staging, have been described previously (Table 1), suggesting the involvement of lymphocytes in the progression of NAFL toward NASH [77]. Accordingly, fibrosis staging, measured by fibroscan, and intrahepatic T-lymphocyte frequency were also found to be positively correlated. In contrast, no correlation was found between fibrosis and circulating T-lymphocyte counts (Table 1) [20], likely due to similar peripheral blood lymphocyte fraction or counts in NAFL and NASH compared to controls [63,72,73]. Nevertheless, positive correlations were found between toll-like receptor 9 (TLR9) expression in peripheral CD4+ and CD8+ T cells and clinical and pathological alterations of NAFLD (Table 1) [78]. Interestingly, in the same study, a downregulation of TLR9 was observed in peripheral T cells (both CD4+ and CD8+ cells) and in intrahepatic CD4+ cells of patients with NAFL, compared to control subjects. Of note, TLR9 is involved in interferon-γ (IFNγ) production by T-lymphocytes, a contributing cytokine to liver injury and inflammation (reviewed in Section 5.3). This finding may suggest a protective adaptation to hepatocellular injury in patients with simple steatosis. On the other hand, a recovery of TLR9 expression was observed in patients with NASH [78]. Given the complex role of IFNγ (see Section 5.3), this observation could be the consequence of a failure of this protective mechanism, contributing to a proinflammatory milieu, or a protective mechanism against liver fibrosis in these patients (Figure 1).

In addition to TLR9, other TLRs, such as TLR2, TLR4 and TLR5 have proven roles in the pathogenesis of NAFLD and its progression to NASH (reviewed in [79]). For example, TLR4, mRNA is overexpressed in the liver of patients with NASH compared to patients with NAFL [80], and TLR4 deficiency in ob/ob mice protects the liver from damage and hepatitis, but not from steatosis [81]. Although TLRs are important IFNγ-regulating factors, the generation and release of this proinflammatory cytokine is also regulated by other mediators, such as substances secreted by different enteric bacteria in NAFLD [79].

Thus, more comprehensive studies should be performed to better understand the role of the different lymphocyte subpopulations in both NAFL and NASH. The main findings regarding the different lymphocyte subsets in this field are described below.

#### 4.3.1. Th1 Cells

Th1 lymphocytes are proinflammatory cells characterized by the production of IFNγ, IL-2 and TNFα. The main cytokines involved in the differentiation of Th0 (naïve T cell) toward the Th1 phenotype are IL-12 and IFNγ, through the activation of signal transducers and activators of transcription (STAT) 1 and STAT4. These cells play an important role in the cellular component of the adaptive immune system (cell-mediated immunity), especially in the host defense against intracellular pathogens, through macrophage activation [32].

Regarding Th1 frequency, there is evidence of a greater peripheral percentage of IFNγ-producing CD4+ cells in NAFL and NASH than in control subjects [78,82,83,84,85]. Despite these findings, no significant correlations were detected between the peripheral percentage of IFNγ-producing CD4+ cells and histological features of NASH, such as steatosis, lobular inflammation, ballooning and fibrosis stage [83].

Regarding intrahepatic examination, Rau et al. documented higher percentages of IFNγ-producing CD4+ cells in the liver, compared to circulating values, in both NAFL and NASH [82]. These results could indicate an enhanced liver infiltration of Th1 cells in these hepatic complications. Nevertheless, determinations in control subjects are required in order to provide proof. Liver upregulation of genes associated with the promotion of Th1 phenotype was described in NASH patients compared to those with NAFLD or obese patients [86].

Despite the above-mentioned results, human studies regarding Th1 cells in NAFLD remain, to date, insufficient sources from which to draw accurate conclusions.

#### 4.3.2. Th2 Cells

Th2 cell differentiation from naïve T cells is mainly driven by IL-2 and IL-4. Functionally, Th2 lymphocytes are involved in host defenses against parasitic infections and play a prominent role in the pathogenesis of allergic diseases. These cells produce several cytokines, including IL-4, IL-5, IL-10 and IL-13, most of them relevant to the orchestration of humoral immunity, through the activation of STAT5 and STAT6 [32].

Very few human studies have investigated the role of this lymphocyte subtype in both NAFL and NASH, and the findings of these remain controversial. While some studies did not find differences in the percentage of IL-4-producing CD4+ cells (considered as Th2 cells) in the peripheral blood of NASH patients compared to control subjects [83,85], other studies found a greater percentage of these circulating cells in both NAFL and NASH patients compared to healthy volunteers [20,82]. Although no differences were observed between NAFL and NASH patients, Rau et al. documented a higher circulating Th2/Treg ratio in patients with NASH compared to those with NAFLD, which was significantly reduced 1 year after bariatric surgery in these NASH patients [82].

Regarding the Th2 cell frequency in liver tissue, a greater percentage of IL-4-producing CD4+ cells was described, compared to circulating values in both NAFL and NASH patients [73]. Although this observation could indicate an increased liver infiltration of Th2 cells, existing data from control subjects are insufficient to prove it.

Given the increased circulating Th2 cell counts in NAFLD patients documented by some authors, along with the apparent enhanced Th2 cell liver infiltration, it seems that this T cell subset likely plays a yet unknown role in NAFLD development and/or progression to NASH.

#### 4.3.3. Th17 Cells

The differentiation of naïve T cells into Th17 is orchestrated by several cytokines, including TGF-β, IL-1β, IL-6, IL-21, IL-23 and TNFα, through the activation of STAT3. These proinflammatory cells are relevant in cell-mediated immunity, especially for the host defense against extracellular pathogens, through the generation and release of proinflammatory cytokines such as IL-17 (mainly the isoforms IL-17A and IL-17F), IL-22 and IL-23 [32]. These cytokines are responsible for the synthesis of some neutrophil chemoattractant chemokines, such as GROα/CXCL1, GROβ/CXCL2 or IL-8/CXCL8 [87].

In the context of NAFLD, IL-17 seems to be the most relevant Th17-related cytokine. IL-17 exacerbates liver inflammation by increasing leukocyte infiltration, stimulating the generation of other proinflammatory mediators and promoting profibrotic effects (reviewed in [66]). The strong proinflammatory response induced by IL-17 is due to the ubiquitous expression of its counterreceptor (IL-17r), which is localized on endothelial and epithelial cells, as well as on monocytes and macrophages [32].

Th17 cells have been studied more extensively in these hepatic complications, but again, findings are not exempt from discrepancies. Whereas Wang et al. described a greater percentage of IL-17-producing CD4+ cells (considered as Th17 cells) in the peripheral blood of NASH patients compared to NAFL or healthy controls [84], Rau et al. did not find any differences [82]. These dissimilarities could have been reliant on the differential diagnostic approaches employed. NAFL and NASH were diagnosed accurately via liver biopsy and liver functional tests by Wang et al. [84], while a noninvasive sonographic NASH score was used by Rau et al. [82]. Nevertheless, the latter study reported a higher circulating Th17/Treg ratio in NASH patients compared to NAFL or healthy controls, which was significantly reduced 1 year after bariatric surgery of NASH patients [82].

Moreover, a greater percentage of hepatic IL-17-producing CD4+ cells was found in both NAFL and NASH patients compared to circulating values [73]. In agreement with this observation, Tang et al. reported a greater IL-17+ cell infiltration in liver tissue of NASH patients compared to control subjects, using immunohistochemical analyses, and also described increased liver mRNA expression of IL-17 and other Th17-related cytokines, such as IL-21 and IL-23 [88]. Liver biopsies from NASH patients presented a higher percentage of Th17 cells and Th17/Treg ratios than those from NAFL subjects [82]. Taken together, these results suggested that Th17 cells, through IL-17 signaling, could be crucial for NAFL progression to NASH.

In addition, circulating levels of IL-17 were found to be increased in NASH patients compared to control subjects. Interestingly, IL-17 levels were significantly higher in those NASH patients with fibrosis than in non-fibrotic patients [89]. These results emphasized the potential link between the IL-17 axis and TGF-β signaling in NASH [66].

Therefore, Th17 cells, through the activation of the IL-17 axis, could play a pivotal role in this liver disorder, especially in the progression of NAFL to NASH and its further complications.

#### 4.3.4. Treg Cells

According to their origin, circulating Treg cells can be divided into two main subsets: those matured in the thymus (tTreg) and those differentiated from naïve T cells, mainly induced by the action of TGF-β (iTreg). Both subsets exert immunoregulatory roles on Th1 and Th17 responses generating anti-inflammatory effects. Treg cells produce important cytokines for immunoregulation, including TGF-β, which amplifies the Treg differentiation from naïve T cells, and the anti-inflammatory IL-10 [32]. However, it is important to highlight that TGF-β, along with IL-6, is also responsible for Th17 cell differentiation from T naïve cells, and possesses profibrotic properties, reflecting the complexity of Th17/Treg balance [32].

Regarding the frequency and role of Treg cells in NAFLD, very few human studies are currently available. According to the scientific literature, and as mentioned above, unbalances in Th1/Treg, Th2/Treg and Th17/Treg ratios were observed in these hepatic complications [82]. In particular, at the circulating level, while both Th1/Treg and Th2/Treg ratios were significantly higher in NAFL or NASH patients than in control subjects, Th17/Treg ratio was only greater in NASH patients, as compared to both NAFL and control subjects. Within the liver, only the Th17/Treg ratio was higher in NASH patients, compared to NAFL patients [82]. Of note, the percentage of circulating CD25++CD4+ cells (considered by the authors as Treg cells) was significantly higher in NASH patients 1 year after bariatric surgery, which was accompanied by a reduction in both circulating Th2/Treg and Th17/Treg ratios [82].

Likewise, a decreased circulating IL-10/IL-17 ratio was detected in NASH patients, compared to non-NASH obese patients [90]. Overall, these findings suggested a link between NAFL development and its progression to NASH with an unfavorable Treg cell balance (decreased Treg cell fraction and/or increased Th1/2/17 frequencies).

#### 4.3.5. CD8+ T Cells

CD8+ T lymphocytes, commonly known as cytotoxic T cells, are formed in the bone marrow and matured in the thymus. They are pivotal players in the elimination of infected or tumoral cells, through the recognition of antigens presented by the major histocompatibility complex (MHC) class I. These cytotoxic effects are achieved through the secretion of cytokines (e.g., IFNγ), cytotoxic agents (e.g., perforin and granzyme) and direct cell-cell contact [32].

Given their effects against tumoral cells, the role of CD8+ cells in NASH-related liver cancer has been extensively studied [91,92]. However, their role in the development of NAFLD and its progression to non-cancerous NASH is limited. According to the current literature, NASH patients and control individuals present similar circulating CD8+ cell counts [82]. However, a greater percentage of IFNγ-producing CD8+ cells was observed in the peripheral blood of NASH patients, compared to control subjects [78,82,85].

As previously outlined, positive correlations were found between the expression of TLR9 in circulating CD8+ T cells and clinical features of NAFLD, such as BMI, TG levels and liver transaminase levels [78]. The observed downregulation of TLR9 in CD8+ cells of patients with NAFL, compared to control subjects, could imply a protective response to hepatocellular injury, since TLR9 is involved in IFNγ production by CD8+ cells [78]. However, no differences were found between NASH patients and control subjects [78], possibly due to a failure in this protective mechanism (contributing to a proinflammatory milieu) or to a protective mechanism against liver fibrosis in these patients. In agreement with these observations, Inzaugarat et al. found no significant correlations between the increased percentages of IFNγ-producing CD8+ cells and histological features of NASH (Table 1) [83].

In contrast, the expression of CD69 (a well-known marker of early lymphocyte activation) was found to be upregulated in peripheral CD8+ cells of patients with NASH, compared to NAFL subjects, suggesting that CD8+ cell activation may be involved in the progression of NAFL to NASH [78]. Despite these findings, little is known regarding the liver infiltration of CD8+ cells in these metabolic disorders in humans and further studies are required.

#### 4.3.6. Natural Killer (NK) and Natural Killer T (NKT) Cells

In addition to T and B cells, there is another group of lymphocytes, the so-called natural killer (NK) cells. Given their crucial role in the elimination of aberrant cells, they are considered relevant cellular players in the innate immune system. NK cell-related cytotoxic activity is due to the release of cytotoxic mediators (e.g., granzyme and perforin) from their granules, as well as the generation of IFNγ, which enhances both NK and CD8+ cell activities against tumor cells [93].

In addition to their role in the immune surveillance of abnormal cells, the role of these cells on NAFL or NASH pathogenesis has been addressed in humans [20,93,94]. Nevertheless, inconsistent results were obtained, likely due to the complexity of liver disease and heterogeneity among NK cells. While some studies described a reduced circulating NK cell (both CD56^bright^ and CD56^dim^ subsets) frequency in NAFLD patients, compared to control subjects [20,93], Stiglund et al. did not find any significant differences [94]. Additionally, the augmented frequency of highly dysfunctional Siglec7^−^CD57^+^PD-1^+^CD56^dim^ NK cell subsets in NAFLD patient bloodstreams demonstrated a functional impairment of NK cells in this context [93]. These studies analyzed liver biopsies for NAFLD diagnosis. However, the observed discrepancies were likely derived from differences in NAFLD patients’ stratification. While Diedrich et al. only compared NAFLD patients with control subjects [20], Sakamoto et al. analyzed results between patients with no or mild fibrosis vs. patients with advanced fibrosis [93], and Stiglund et al. analyzed 3 groups: NASH patients vs. NAFL patients vs. control subjects [94].

In the analysis of liver biopsies, a greater percentage of CD56^dim^ NK cells was observed in NAFLD patients, which could account for their decreased percentage in circulation [20]. Nonetheless, the same study revealed a negative correlation between total NK cell frequency in the liver and the fibrosis stage measured by fibroscan elastography (Table 1) [20].

NKG2D and CD69 are markers of NK cell activation. In this context, while some studies documented higher peripheral and liver NK cell activation in NAFL/NASH patients [93,94], Diedrich et al. described the opposite in circulation [20].

On the other hand, there is a CD3+ T-lymphocyte subset (CD3 is not expressed by NK cells) that expresses some NK cell markers (CD56 or CD161), the so-called NKT cells. In NAFLD, a greater proportion of NKT cells in both blood and liver was detected, which seemed to be associated with the severity of the disease [95,96]. Again, there have been few (and contradictory) results regarding NKT cell activation. Indeed, both greater and reduced circulating NKT cell activation have been associated with this complex disease [20,97]. The frequency of circulating NKG2D+ NKT cells was negatively correlated with the grade of steatosis (Table 1) [20].

Similarly, although several attempts to investigate the role of NK/NKT cells in NAFLD have been carried out, their role in this disease has remained elusive.

## 5. Relevant Soluble Mediators in NAFLD

### 5.1. High-Sensitivity C-Reactive Protein (hs-CRP)

High-sensitivity C-reactive protein (hs-CRP) has traditionally been considered a sensitive systemic marker of inflammation and cardiovascular risk [98]. This protein is mainly overproduced by the liver in response to proinflammatory cytokines, such as IL-1β or TNFα. In NAFLD patients, higher hs-CRP circulating levels were consistently verified, compared to control subjects [99,100]. Accordingly, a recent meta-analysis described a significant positive association between circulating CRP levels and, not only NAFLD risk, but also NASH and liver fibrosis [101]. In fact, follow-up studies pointed out serum hs-CRP as a predictive biomarker of NAFLD [102,103]. Furthermore, some observations have associated the hs-CRP levels with the severity of fatty liver [104] and the fibrosis stage [105]. Notwithstanding, in obese patients, Zimmermann et al. found that hs-CRP correlation with steatosis was BMI-independent while, in NASH patients, it was BMI-dependent (Table 1) [106]. Therefore, hs-CRP could be a biomarker of steatosis but not of NAFLD severity in obese patients [106].

Hs-CRP was also reported as an important factor connecting NAFLD and cardiovascular diseases. Recently published 5–10-year follow-up studies indicated that elevated hs-CRP levels were related to increased risk of ischemic stroke [107,108] and cardiovascular mortality [109] among NAFLD patients. In fact, a 4-year follow-up study indicated that NAFLD patients with higher circulating levels of hs-CRP showed a higher risk of developing coronary artery calcification [110]. In this line, hs-CRP induces endothelial expression of CAMs, including the intercellular adhesion molecule-1 (ICAM-1) and the vascular cell adhesion molecule-1 (VCAM-1), as well as the generation and the secretion of chemokines, such as IL-8/CXCL8, MCP-1/CCL2, RANTES/CCL5, fractalkine/CX_3_CL1 or CXCL16, through NF-κB activation, leading to leukocyte-endothelium interactions, an early key step of atherogenesis and related complications [54,111]. Altogether, these data indicate that increased circulating levels of hs-CRP are an independent predictive factor of poor prognosis in NAFLD patients.

### 5.2. Tumor Necrosis Factor-α (TNFα)

TNFα is a pivotal proinflammatory cytokine which, in the liver, is mainly produced by Kupffer cells [112]. It is also produced by other immune players, such as Th1 lymphocytes, neutrophils, NK cells, monocytes, or even endothelial cells and hepatocytes [18,20,113]. It exerts its activity through an interaction with its counterreceptors: TNFR1, ubiquitously expressed, and TNFR2, essentially restricted to hematopoietic and endothelial cells [25]. The activation of these receptors triggers the activation of several signaling pathways (NF-kB, Mitogen-Activated Protein Kinases (MAPK) and caspase-8), leading to systemic inflammation and an acute phase reaction that stimulates hepatocytes to generate the neutrophil-related chemokines IL-8/CXCL8 and growth-regulated protein-α (GROα/CXCL1), thereby promoting the progression of NAFLD [18,113].

Higher plasma levels of TNFα have been described in patients with NAFLD compared to control subjects [74], in agreement with two recent meta-analyses [101,114]. TNFα levels were also significantly associated with both NASH and liver fibrosis [101]. Indeed, elevated circulating levels of this proinflammatory cytokine, and also high levels of the soluble form of TNFR1, have been associated with NAFLD severity [25,114]. However, differences in circulating TNFα levels between NAFL and NASH conditions remain unclear; while some studies did not find significant differences [115,116], Jarrar et al. did [117]. In addition, NASH patients were shown to overexpress TNFα in both liver and adipose tissue, and enhanced expression of both TNFα and TNFR1 were associated with advanced liver fibrosis [118].

Hence, it could be tempting to suggest that the TNFα/TNFR1 axis plays a role in the progression of NAFL toward NASH. However, though some beneficial effects on glucose homeostasis were detected in a TNFR1-deficient murine model of diet-induced NASH, no hepatic improvement was observed by Bluemel et al. [119]. It is worthy of mention that TNFR1 had only been knocked-out in hepatocytes, which could have been a limitation of this research. Indeed, other authors described a reduction of hepatocellular injury, as well as liver steatosis and fibrosis, in a diet-induced NASH murine model after an 8-week treatment with an anti-TNFR1 antibody [24]. The differences found between these two reports could have been reliant on the TNFR1 inhibition approach (hepatic vs. systemic inhibition), suggesting that extra-hepatic players might also have an important role in NAFLD development.

Anti-TNFα therapy is currently used to control immune-related diseases (e.g., inflammatory bowel disease, psoriatic arthritis or rheumatoid arthritis). A recent retrospective cohort study focused on the effects of anti-TNFα therapy against NAFLD development [120]. The analysis showed no beneficial effects of this treatment in this metabolic context. However, it is relevant to highlight the short-term follow-up (1.5 years) and that immune-related diseases were present, which could have compromised the eventual beneficial hepatic effects.

Taking all this into account, TNFα/TNFR1 axis blockage emerges as a promising approach in the treatment of NAFLD. However, as far as we are aware, no clinical trials are currently available regarding anti-TNFα/TNFR1 therapy in this pathology.

### 5.3. Interferon Gamma (IFNγ)

IFNγ is a proinflammatory cytokine predominantly secreted by several immune cells, including Th1 cells, cytotoxic T lymphocytes, NK and NKT cells [32,85,93,97]. This cytokine participates in both innate and adaptive cellular immune responses against intracellular pathogens, as well as in antitumoral activity. IFNγ actions rely on its interaction with the surface heterodimeric receptors IFNγR1 and IFNγR2, which activates the Janus kinase (JAK) and the Signal Transducer and Activator of Transcription-1 (STAT1) pathway [113].

In NASH, IFNγ has been described to be involved in macrophage polarization toward a proinflammatory M1-like phenotype, to the detriment of an anti-inflammatory M2-like phenotype [121]. While IFNγ contributes to liver injury and inflammation, this cytokine also exerts a protective effect against liver fibrosis, through the STAT1 signaling pathway [113]. Accordingly, the depletion of IFNγ in obese mice accelerated the development of liver fibrosis [122]. This anti-fibrotic action of IFNγ seemed to be related to the decreased M2-phenotype macrophage polarization, since these cells led to an overproduction of profibrotic mediators (reviewed in [123]).

NK cell activation also seems to be involved in this antifibrotic response, by inhibiting the activation of HSCs and the activity of profibrotic mediators, such as TGF-β and IL-13 [113]. Along these lines, Tosello-Trampont et al. demonstrated that NKp46+ NK cells (IFNγ-producing cells) prevented the development of liver fibrosis through an IFNγ-mediated M1-phenotype polarization [121].

Despite these findings, evidence was provided from diet-induced murine NASH models indicating that IFNγ- or Stimulator of interferon genes (STING)-deficiency attenuates steatosis and liver fibrosis, as well as TNFα and IL-6 mRNA liver expression [124,125]. These contradictory findings could have been reliant on inherent differences in NASH-induced mice models and on the difficulty to simulate human-like NASH in this species.

In humans, circulating IFNγ levels were found to be significantly elevated in patients with NAFLD compared to control subjects [77]. Moreover, serum levels of this cytokine were found to be positively correlated, not only with the number and size of hepatic lymphocyte aggregates, but also with the severity of fibrosis in these patients (Table 1) [77]. The latest correlation could demonstrate a pathogenic role of IFNγ in fibrogenesis or, on the contrary, a physiological response in the attempt to reverse the development of liver fibrosis. Of note, a randomized controlled trial demonstrated fibrosis scores’ improvement in patients with chronic hepatitis B virus infection, after a 9-month treatment with IFNγ [126]. Although both hepatic diseases were associated with fibrosis, these findings could not simply be extrapolated to a NASH context, given the substantial differences in the etiopathogenesis.

### 5.4. Monocyte Chemoattractant Protein-1 (MCP-1/CCL2)

Monocyte chemoattractant protein-1 (MCP-1/CCL2) is a proinflammatory chemokine mainly produced by monocytes and macrophages. However, other cells also express it to a lesser extent, such as endothelial cells, smooth muscle cells and HSCs [18,54]. This chemokine attracts cells expressing its counterreceptor CCR2, particularly monocytes and macrophages [18].

According to the current literature, the MCP-1/CCR2 axis seems to play a role in NAFLD development and progression. While increased CCR2 mRNA expression was found in the liver of NASH patients compared to NAFL patients or control subjects [127], liver mRNA expression of its counterligand MCP-1/CCL2 was positively correlated with liver fat content (Table 1) [128,129,130]. Moreover, Liu et al. claimed that MCP-1/CCL2 was a relevant player in insulin resistance, steatosis, liver inflammation and fibrosis development [130].

Although enhanced MCP-1/CCR2 axis expression in the liver seems clear, there have been some discrepancies relative to the circulating levels of MCP-1/CCL2 in NAFLD. Several studies observed higher circulating levels of this chemokine in NAFLD patients compared to control subjects, being significantly higher in NASH than in NAFL patients [127,131,132]. In addition, Puengel et al. found positive correlations between circulating levels of this chemokine and serum levels of gamma-glutamyl transferase and fibroblast growth factor 21, as well as FIB-4 score (Table 1), suggesting a link between MCP-1/CCL2 and both liver damage and fibrosis [133]. In contrast, the same authors found no associations between the circulating levels of this chemokine and steatosis or lobular inflammation [133]. These observations were in agreement with Ali et al., who suggested MCP-1/CCL2 as a potential biomarker to discriminate NAFL from NASH [132]. Despite these findings, other studies reported no differences regarding circulating levels of MCP-1/CCL2 between NASH, NAFL and control subjects [116,133]. Indeed, no association between circulating levels of MCP-1/CCL2 and NAFLD has been found in a recent meta-analysis [101]. These discrepancies were certainly due to the heterogenicity of NAFLD, along with the limited number of patients recruited in these studies.

In summation, there has been increased interest in anti-MCP-1/CCR2 axis therapy in NAFLD. In murine models of metabolic disorders, the pharmacological inhibition of this axis was shown to exert beneficial effects on obesity, insulin resistance, steatosis and inflammation [18]. Moreover, Cenicriviroc, a dual CCR2/CCR5 antagonist, reduced fibrosis after one year of treatment in a phase II clinical trial in NASH patients [75]. Nevertheless, clinical studies on this drug were interrupted in phase III, due to a lack of effect over two years of therapy [133]. Therefore, MCP-1/CCL2-related evidence in NAFLD is still controversial, suggesting that MCP-1/CCL2 is not a single pivotal player in this disease and certainly depends on other factors.

### 5.5. Transforming Growth Factor-β (TGF-β)

TGF-β is a cytokine with immunosuppressive, anti-inflammatory and profibrotic properties secreted by several cells, including HSCs, epithelial cells and immune cells, especially Treg lymphocytes [18,32]. In addition to its crucial role in naïve T cell differentiation towards the Th17 and Treg phenotype, TGF-β has been described to be the most potent fibrogenic cytokine [32,134]. In fact, this cytokine is a key driver in HSC activation, leading to excessive production of extracellular matrix, which in turn induces the differentiation of HSCs into collagen-producing myofibroblasts [18,134].

In the liver, mRNA expression of the most common isoform of TGF-β (TGF-β1) was increased in a murine model prone to develop liver fibrosis (obese IFNγ-/- subjected to a high-fat diet) [122]. Both TGF-β blockade and TGF-β receptor-deficiency partially reduced liver fibrosis in mice [122,135], suggesting that, though TGF-β exerted a key role in fibrosis, other mediators were orchestrating the process as well. Indeed, Hart et al. also suggested IL-13 as a profibrotic cytokine that collaborated with TGF-β in fibrosis development [122].

Apart from antifibrotic effects, the anti-TGF-β therapy did not alter the grade of steatosis (determined by hepatic TG content measurement) in NASH murine models [122]. Accordingly, Hasegawa et al. detected increased TGF-β1 plasma levels in patients with NASH, compared to patients with NAFL or control subjects [136]. Moreover, a higher TGF-β signaling activation was also reported in the liver of NASH patients with fibrosis, when compared to patients with mild NAFLD without fibrosis [135].

All these findings strengthen the evidence that TGF-β is pivotal for fibrosis, but not for steatosis development. Regarding this, circulating levels of this cytokine might be a useful biomarker to distinguish NASH from NAFL, and anti-TGF-β therapy could be a promising approach to alleviate liver fibrosis.

### 5.6. Interleukins (ILs)

Interleukins (ILs) are a family of cytokines produced by several cells with specific immunomodulatory functions, especially in cell-mediated immunity. Among these, some are proinflammatory, while others are anti-inflammatory. During the last few years, some ILs have been associated with NAFL and NASH.

Among the IL-1 superfamily of cytokines, produced by both immune and nonhematopoietic cells, IL-1β and IL-18 have been particularly related to NAFLD. Both cytokines need to be activated by NLRP3 inflammasome to exert their functions [113]. IL-1β is barely expressed in healthy livers; however, in NASH, IL-1β expression is enhanced, mainly due to macrophage and necroptotic hepatocyte secretion [42,113]. This proinflammatory cytokine was found to promote steatosis (by stimulating TG synthesis and cholesterol accumulation in the liver), inflammation (by upregulating ICAM-1 and inducing IL-6 and TNFα expression) and fibrosis (by activating HSCs and inducing the production of profibrogenic factors) [137]. According to a recent meta-analysis, IL-1β levels were found to be significantly associated with hepatic fibrosis but not with steatosis [101]. While IL-1β triggered proinflammatory responses, IL-18 (mostly produced by macrophages in the liver) exerted pro- or anti-inflammatory effects depending on the environment [113]. In liver steatosis, IL-18 displayed some protective effects in murine models. IL-18 deficiency caused dyslipidemia and steatosis, while the administration of a recombinant IL-18 reverted this process [138]. Despite its beneficial effects in reducing hepatic lipid storage, IL-18 was reported to participate in the development of fibrosis [139]. Indeed, Hohenester et al. positioned IL-18, rather than IL-1β, as a pivotal player for liver injury in NAFLD murine model [140]. In humans, IL-18 circulating levels were found to be significantly increased in patients with NASH or cirrhosis, compared to control subjects [141,142]. Furthermore, several studies reported positive correlations between circulating IL-18 levels and inflammation, liver injury, fibrosis or the severity of the disease (Table 1), positioning IL-18 as a reliable biomarker to predict NASH diagnosis [141,142,143]. As mentioned by Somm et al. [139], IL-18 has exhibited interesting beneficial effects against steatosis—however, an overexpression of this cytokine might be deleterious for liver integrity.

IL-4 and IL-13 are two closely-related cytokines secreted by several cells, including Th2 cells, granulocytes and monocyte/macrophages. They participate in type 2 immune response [113]. In liver injury, IL-13 signaling activates the secretion of the eosinophil-chemoattractant eotaxin-1/CCL11 from epithelial cells and fibroblasts. On the other hand, recruited eosinophils have been described to secrete IL-4 in an attempt to promote liver regeneration [144]. Both ILs seem to be involved in this process, given their role in relevant growth factors secretion from M2 macrophages, such as fibroblast growth factor (FGF), connective tissue growth factor (CTGF) and platelet-derived growth factor (PDGF) [113]. This response promotes collagen secretion from fibroblasts to repair damaged tissue, although an overactivation of this response can lead to fibrosis [113]. Indeed, as mentioned above, IL-13 collaborates with TGF-β in fibrosis development, given that IL-13 expression is increased in NASH murine model, and the hepatic beneficial effects of IL-13/TGF-β blockade are more evident than TGF-β blockade alone [122]. Furthermore, IL-5 is another type 2 cytokine with eosinophilic properties. It is secreted by several cells, including Th2 cells, type 2 innate lymphoid cells (ILC2s), mast cells, and eosinophils. This interleukin participates in eosinophil differentiation, degranulation, recruitment and adhesion [113]. In this line, Hart et al. found higher production of IL-4, IL-5 and IL-13 from intrahepatic lymphocytes of NASH patients, compared to healthy subjects, suggesting a potentially relevant role of type 2 cytokines in NASH progression [122].

IL-6 is mainly produced by immune cells (including Th17 lymphocytes), although it can also be released by hepatocytes, among other cells [18]. Its action in liver disease is very complex, since it seems to act as both a pro- and an anti-inflammatory cytokine. On one hand, there is evidence that IL-6 exerts a hepatoprotective effect against liver steatosis due to the reduction of ROS generation and the subsequent oxidative stress attenuation [145]. In addition, a recent Mendelian randomization study suggested that IL-6 receptor blockade might increase the risk of NAFLD [146], which strengthens the potential protective role of IL-6 in this disorder. However, Yamaguchi et al. described a paradoxical role of IL-6 in NAFLD. While IL-6 signaling blockade resulted in accelerated liver steatosis in a diet-induced NASH murine model, it also improved liver injury [147]. Accordingly, IL-6-deficient mice showed attenuated NASH, as compared to controls [148]. These effects were due to the complex signaling of this interleukin. IL-6 not only induced STAT3 activation, leading to inflammation and liver damage, but also decreased the expression of sterol regulatory element binding protein-1 (SREBP1), alleviating steatosis [147]. In addition, the synthesis of several acute-phase proteins was orchestrated by IL-6, including CRP, contributing to the deleterious effects of this cytokine [18]. In humans, evidence of enhanced IL-6 serum levels has been detected in both NAFL and NASH patients [131,149], as well as increased IL-6 hepatic expression in NASH which seemed to correlate with the severity of the disease (Table 1) [150]. Finally, in a recent meta-analysis and an observational study, IL-6 levels were found to be associated with both increased risks of NAFLD [101] and atherosclerosis development in these subjects [99].

IL-8, also known as CXCL8, is one of the most potent neutrophil chemoattractant cytokines. It is only expressed in humans. Given the evidence of altered neutrophil activity in NAFLD (described in Section 4.1), the measurement of circulating levels of this chemokine has been addressed in several studies. Evidence of increased circulating levels of IL-8 in patients with NASH, compared to NAFL patients [63,115] or healthy subjects, has been found [63,116]. However, while Jarrar et al. found differences between NAFLD patients and obese controls, the authors did not observe differences between NAFL and NASH patients [117], as described in an Auguet et al. study [116]. These discrepancies between NAFL vs. NASH findings were certainly due to the heterogenicity of these pathological conditions, along with the limited number of patients recruited in these studies. Nonetheless, from the current scientific literature, we were able to conclude that IL-8 expression is enhanced in NAFLD.

IL-10 is mainly secreted by Treg lymphocytes and M2 macrophages [32,151]; however, some liver-related cells also express this cytokine, including hepatocytes, HSCs and Kupffer cells [18]. Given its activity in inhibiting the functions of several immune cells (including Th1 cells, Th17 cells, monocytes and M1 macrophages), IL-10 has been considered an anti-inflammatory cytokine. Additionally, since it also plays a role in extracellular matrix remodeling, IL-10 has been related to fibrogenesis. Its role in fibrosis development is complex; depending on the nature of the pathology, IL-10 may exert anti- or profibrotic effects (reviewed in [151]). In the liver, IL-10 exerts beneficial effects against fibrosis development, through its secretion by Treg cells [32,151]. Nevertheless, the role of this cytokine in NAFLD has been scarcely investigated. Despite the unfavorable Treg cell balance (described in Section 4.3.4), no significant differences in IL-10 plasma levels were documented between subjects with NAFL, NASH and healthy controls [116,152]. In addition, no significant association was found between circulating levels of this cytokine and NAFLD, according to a recent meta-analysis [101]. In general, IL-10 does not appear to be a relevant peripheral biomarker for identifying neither an increased risk of NAFLD nor for determining NAFLD stages.

Within the IL-17 family, the most abundant form is IL-17A. This is mainly produced by Th17 cells, but also by CD8+ cells, NKT cells, NK cells and ILCs [113]. IL-17A seems to play a deleterious role in NAFLD development and progression. On one hand, it stimulates the synthesis of two relevant neutrophil chemoattractant cytokines—IL-8/CXCL8 and GROα/CXCL1—that contribute to neutrophil infiltration in the liver and amplify liver damage. On the other hand, IL-17A activates macrophages and leads them to produce proinflammatory and profibrotic cytokines, such as IL-1β, IL-6, TNFα and TGF-β. In fact, the blockade of IL-17 signaling has been shown to attenuate systemic inflammation, hepatic steatosis and fibrosis in experimental models of NASH (reviewed in [113]). In humans, while hepatic IL-17A mRNA was positively correlated with hepatic TG content (Table 1) [153], no differences in IL-17 plasma levels were found between NAFL, NASH and healthy controls [116]. So far, the expression of this cytokine in NAFLD has been poorly explored.

**Table 1 ijms-24-02313-t001:** Correlations between NAFLD features and cellular or soluble markers.

NAFLD Features	Correlation	Marker	References
Liver inflammation	+	MPV	[48]
	+	Intrahepatic T-lymphocyte frequency and aggregates	[77]
	+	Circulating IL-18 levels	[142]
	+	Hepatic IL-6 expression	[150]
	NR	Peripheral percentage of IFNγ-producing T cells	[83]
	NR	Circulating MCP-1/CCL2 levels	[133]
Steatosis	+	MPV	[48,49]
	+	Circulating hs-CRP levels	[101,104,106]
	+	Hepatic IL-17A mRNA expression	[153]
	+	Hepatic MCP-1/CCL2 mRNA expression	[128,129,130]
	−	Peripheral activated (NKG2D+) NKT cell frequency	[20]
	NR	Peripheral percentage of IFNγ-producing T cells	[83]
	NR	Circulating MCP-1/CCL2 levels	[133]
Fibrosis	+	MPV	[48]
	+	Neutrophil/lymphocyte ratio (NLR)	[48,57,63,66,67]
	+	Intrahepatic T-lymphocyte frequency and aggregates	[77]
	+	Circulating IFNγ levels	[77]
	+	Circulating MCP-1/CCL2 levels	[133]
	+	Circulating IL-18 levels	[143]
	+	Circulating hs-CRP levels	[101,105]
	+	Hepatic IL-6 expression	[150]
	−	Hepatic NK cell frequency	[20]
	NR	Peripheral T-lymphocyte counts	[20]
	NR	Peripheral percentage of IFNγ-producing T cells	[83]
Liver injury	+	MPV	[48]
	+	TLR9 expression on circulating CD4+ or CD8+ cells	[78]
	+	Circulating MCP-1/CCL2 levels	[133]
	+	Circulating IL-18 levels	[142]
	NR	Peripheral percentage of IFNγ-producing T cells	[83]
Body mass index	+	Hepatic MPO mRNA expression	[61]
	+	TLR9 expression on circulating CD4+ or CD8+ cells	[78]
Dysglycemia	+	Hepatic MPO mRNA expression	[61]
	+	Hepatic IL-6 expression	[150]
Hypertriglyceridemia	+	TLR9 expression on circulating CD8+ cells	[78]

(+) positive correlation; (−) negative correlation; (NR) not related. Abbreviations: hs-CRP, high-sensitivity C-reactive protein; IFNγ, interferon-γ; MCP-1, monocyte chemoattractant protein-1; MPO, myeloperoxidase; MPV, mean platelet volume; TLR9, toll-like receptor 9.

## 6. Concluding Remarks

As detailed in the present review, NAFLD development and progression requires several cellular and soluble players, some with paradoxical or contradictory effects, which hampers the comprehension of the full molecular insight of this complex pathology. In addition, NAFLD is a multifactorial disease, often associated with other inflammatory or cardiometabolic disorders, leading to a wide heterogeneity of clinical features. Thus, discrepancies among human studies in this field may also be due to the different diagnostic approaches employed. While liver biopsy remains the gold standard to establish NAFL/NASH diagnosis, several studies opted for noninvasive strategies, which could have led to differences in NAFLD stratification. Moreover, research in this field has also been challenged by several difficulties to simulate human-like NAFLD in mice, owing to inherent physiological differences between species.

Nevertheless, some conclusions may be established from the present literature review. For example, NAFLD-related thrombocytopenia is due, not only to a reduction in TPO production by liver injury, but also to an increase of both platelet aggregation and infiltration in the liver of these patients. These observations are more closely associated with liver inflammation and fibrosis rather than steatosis, suggesting that platelets play a more relevant role in NAFL progression toward NASH, rather than NAFL development itself.

Aside from platelets, several leukocyte subsets and related cytokines were reported to contribute to NAFLD development (Figure 1). Neutrophils seemed to be involved in fibrogenesis, promoting the production of profibrotic mediators, such as TGF-β and α-SMA. This explained the positive correlations documented between NLR and both NASH severity and liver fibrosis. Of note, neutrophils, along with Th17 cells, are an important source of IL-17. IL-17 not only stimulates the production of proinflammatory and profibrotic, but also leads to an upregulation of important neutrophil chemoattractant cytokines. Therefore, neutrophils appear to amplify their response in NAFL to NASH progression. Since no differences in plasma levels of this interleukin have been reported between NAFLD patients and control subjects thus far, interest in IL-17 as a peripheral biomarker to diagnose or stratify NAFLD has decreased.

Regarding monocytes and the related MCP-1/CCR2 axis, increased circulating monocyte fraction, along with greater MCP-1/CCR2 axis expression in the liver, were found in NAFLD patients. However, conclusions about MCP-1/CCL2 plasma levels remain controversial and further investigation will be required.

Furthermore, no relevant differences in CD4+ or CD8+ cell counts between control subjects, NAFL and NASH patients were described. However, both T-cell infiltration and aggregation in the liver were positively correlated with lobular inflammation and fibrosis staging. In addition, both TLR9 expression in T cells and peripheral/liver NKT cell frequencies were found to be linked to the severity of the disease, which could explain the higher plasma levels of IFNγ in NAFLD subjects. Despite its contribution to liver injury and inflammation, IFNγ seems to exert an antifibrotic activity. However, IFNγ plasma levels were found to be positively correlated to fibrosis staging, likely due to a physiological response to counteract profibrotic mediators’ activity. IFNγ might constitute a potential, but not selective, biomarker to identify NAFLD risk. Additionally, an imbalance among CD4+ cell subsets in this disease was established, favoring both a Th1 and Th17 cell proinflammatory environment over a Treg cell anti-inflammatory condition. Nevertheless, Treg cells, along with HSCs and epithelial cells, are a source of TGF-β, an immunosuppressive cytokine with a potent fibrogenic activity. In addition, patients with NASH present higher TGF-β plasma levels than NAFL or control subjects, which might place TGF-β as a useful peripheral biomarker to distinguish NASH from NAFL.

Other soluble mediators, such as TNFα, IL-1β and IL-6, seem to be pivotal cytokines in the early steps of NAFLD development. First, they amplify the proinflammatory response. Second, and contrary to the great majority of the other soluble mediators studied, they seem to be upregulated in simple steatosis (NAFL). Indeed, out of 19 soluble mediators, only these four above-mentioned have been significantly associated with NAFLD, according to a recent meta-analysis [101]. Hence, these cytokines might be useful biomarkers to evaluate steatosis severity.

Additionally, the detection of type 2 cytokines such as IL-4, IL-5 and IL-13, suggests a potential role of Th2 cells in NASH progression. Particularly, in liver injury, IL-13 seems to trigger a cytokine cascade, promoting the switching of macrophages toward an M2 profibrotic phenotype. Finally, IL-18 is also highlighted by its paradoxical role in NAFLD. While it shows beneficial effects in dyslipidemia and steatosis, IL-18 seems to participate in fibrogenesis. Accordingly, enhanced IL-18 plasma levels were positively correlated with liver inflammation, injury and fibrosis, which might place this interleukin as an additional predictive biomarker of NASH.

Taking all these results together, some research topics were proposed (see Table 2) to contribute to advances in NAFLD early diagnosis and treatment.

In conclusion, it seemed, after review, that most of the mentioned cellular and soluble mediators, with the exception of TNFα, IL-1β, IL-6 and hs-CRP, were more related to inflammation and fibrosis than steatosis itself. This reinforced the idea that, although inflammatory mediators are not crucial for increased fat accumulation in the liver, they play a pivotal role in NAFL to NASH progression. In particular, unresolved hepatic inflammation may lead to liver fibrosis. Herein, promising therapeutic approaches were aroused, such as the TNFα/TNFR1 axis blockage to prevent NAFLD progression, anti-TGF-β therapy to alleviate liver fibrosis or neutralization of IFNγ activity to revert both liver inflammation and injury. Nevertheless, additional efforts should be carried out to further understand the role of the immune system in NAFLD. This would help to improve diagnosis and discover new therapeutic approaches to prevent, halt or revert this complex metabolic disease.

## Figures and Tables

**Figure 1 ijms-24-02313-f001:**
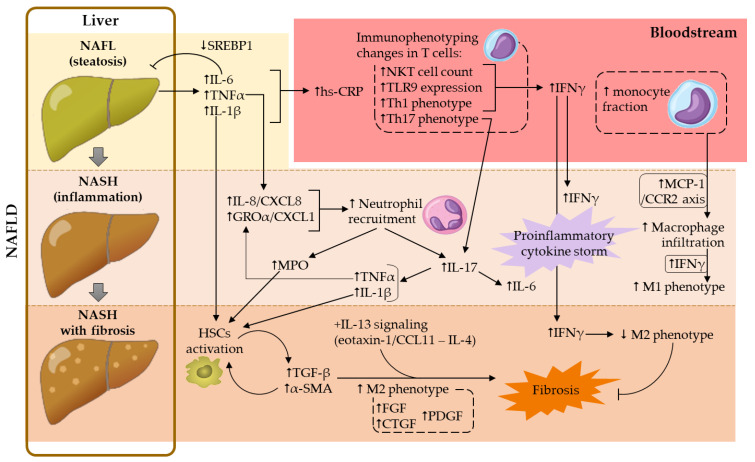
Contribution of the main cellular and soluble mediators studied in NAFLD development. α-SMA, α-smooth muscle actin; CTGF, connective tissue growth factor; FGF, fibroblast growth factor; GROα, growth regulated protein-α; hs-CRP, high-sensitivity C-reactive protein; HSCs, hepatic stellate cells; IFNγ, interferon-γ; IL, interleukin; MCP-1, monocyte chemoattractant protein-1; MPO, myeloperoxidase; NAFL, non-alcoholic fatty liver; NAFLD, non-alcoholic fatty liver disease; NASH, non-alcoholic steatohepatitis; NKT, natural killer T; PDGF, platelet-derived growth factor; SREBP1, sterol regulatory element binding protein-1; TGF-β, transforming growth factor-β; Th, T helper; TLR9, toll-like receptor 9; TNFα, tumor necrosis factor-α.

**Table 2 ijms-24-02313-t002:** Future research topics proposed to improve NAFLD diagnosis and treatment.

Future Research Items to Be Addressed
1. Discover new molecular pathways involved in NAFL development and in its progression to NASH.
2. Clarify the role of each immune player involved in NAFLD and characterize their phenotype.
3. Develop novel noninvasive techniques to diagnose and stratify NAFLD correctly.
4. Collect and analyze longitudinal data to better understand the cellular and molecular changes in NAFLD progression.
5. Identify new biomarkers for NAFLD prediction, diagnosis, stratification or disease reversion.
6. Study novel therapeutic targets to prevent, halt or revert NAFLD.
7. Develop new NAFLD animal models that fully simulate human pathology.

## Data Availability

Not applicable.

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
