# Peer review of "Overview of Cellular and Soluble Mediators in Systemic Inflammation Associated with Non-Alcoholic Fatty Liver Disease"

_ijms, 2023, doi:10.3390/ijms24032313_

Round 1

Reviewer 1 Report

Please see the attached file for comments.

Author Response

RESPONSES TO REFEREE 1 

The authors wish to express their thanks for the expert review of their manuscript. We have endeavored to address all of the points raised by the referee following the same order of the report. To ease the work of the referee, all the modifications are highlighted in yellow in the resubmitted version of the manuscript.

 In the review “Overview of cellular and soluble mediators in the systemic inflammation associated with Non-Alcoholic Fatty Liver Disease”, the Authors focused on the main cellular and soluble mediators in the development of non-alcoholic fatty liver disease, the most prevalent chronic hepatic disease in Western countries.Although the subject is interesting and comprehensively described, a few concerns need to be addressed before the manuscript is ready for publication in the International Journal of Molecular Sciences. 

Comments: 

1. Over the last two decades, many criticisms have been voiced about the nomenclature and definition of non-alcoholic fatty liver disease (NAFLD) in regards not only to the prominent role that alcohol plays in the definition but also on the negative impacts of the nomenclature including trivialization, stigmatization and less consideration of the disease in health policy. Recently, a consensus by an international panel of experts recommended a change in name for NAFLD to metabolic (dysfunction) associated fatty liver disease (MAFLD). This issue is reviewed by Fouad, Y., Waked, I., Bollipo, S., Gomaa, A., Ajlouni, Y., & Attia, D., 2020 (What's in a name? Renaming ‘NAFLD’ to ‘MAFLD’." Liver international 40.6: 1254-1261). For details please see also a paper by Eslam, M., Sanyal, A. J., George, J., Sanyal, A., Neuschwander-Tetri, B., Tiribelli, C., ... & Younossi, Z., 2020 (MAFLD: a consensus-driven proposed nomenclature for metabolic associated fatty liver disease. Gastroenterology, 158(7), 1999-2014). Therefore, I suggest that the Authors mention the recommended nomenclature of the described liver disease. (It should be noted that the acronym “MAFLD” is used in Ref. 107).

Thank you for the suggestion. We mentioned the recently established term MAFLD in lines 54-60 and added it to the keywords. 

2.     I suggest “TGs” instead of “TG” to abbreviate “triglycerides”. Please see line 157 for comparison. 

Thank you for the suggestion, the correction has been made in lines 112 and 113. 

3.     Line 410: The section numbering is incorrect. It should be “4.3. Lymphocytes” instead of “4.2. Lymphocytes” (please see line 375: 4.2. Monocytes).

We apologize for this mistake, all the corrections regarding section numbering have been made. 

4.     Line 461: It should be “4.3.1. Th1 cells” instead of “4.2.1. Th1 cells”.

We apologize for this mistake, all the corrections regarding section numbering have been made. 

5.     Line 484: It should be “4.3.2. Th2 cells” instead of “4.2.2. Th2 cells”.

We apologize for this mistake, all the corrections regarding section numbering have been made. 

6.     Line 508: It should be “4.3.3. Th17 cells” instead of “4.2.3. Th17 cells”.

We apologize for this mistake, all the corrections regarding section numbering have been made. 

7.     Line 549: It should be “4.3.4. Treg cells” instead of “4.2.4. Treg cells”.

We apologize for this mistake, all the corrections regarding section numbering have been made. 

8.     Line 575: It should be “4.3.5. CD8+ T cells” instead of “4.2.5. CD8+ T cells”.

We apologize for this mistake, all the corrections regarding section numbering have been made. 

9.     Line 606: It should be “4.3.6. Natural killer (NK) and natural killer T (NKT) cells” instead of “4.2.6. Natural killer (NK) and natural killer T (NKT) cells”.

We apologize for this mistake, all the corrections regarding section numbering have been made. 

10.  Line 741: I suggest “in this species” Instead of “in this specie”.

Thank you for the suggestion, the correction has been made in line 745. 

11.  Lines 905-906: It should be “Despite the unfavorable Treg cell balance described in section 4.3.4. Treg cells” instead of “Despite the unfavorable Treg cell balance described in section 4.2.4. Treg cells”. Please see comments no. 3-7.

We apologize for this mistake, all the corrections regarding section numbering have been made. 

12.  Line 921: IL-17A mRNA (“IL-17A mRNA in Table 1”). See also line 739.

Thank you for this remark, the corrections have been made in lines 743, 764 and 925. 

13.  Table 1: I suggest to verify references cited for MPV (please see lines 246-247).

All the references for MPV were re-analyzed and some adjustments were done in line 249. 

14.  Line 1014: The acronym for non-alcoholic fatty liver is incorrect. It should be “NAFL” instead of “NALF”.

We apologize for this mistake. The correction has been made (line 1006). 

15.  References: Please see comment no. 1 to extend the bibliography.

The bibliography suggested in comment no. 1 was added (see Ref 6 and 7). 

16.  References: Journal titles should be written in the uniform manner.

All the references were revised and a modification in Ref 20 was made.

Reviewer 2 Report

My comments are as follows:

1.The authors have written Review Article well but it is too long and should have been 10-15 pages(excluding references).

2. Line #167. TLRs have also been associated with the development of NAFLD..... please check original Ref #35, it is mentioned NASH and not NAFLD. The authors should check and correct it.

3. Line#659. The Author should check Reference #104 where in original reference mentioned NASH and not NAFLD. 

4. Line#1014 Author should check the word  NALF (typo error), it should be NAFL to NASH.

5. Concluding remarks should have been in brief.

6.  The authors should have added a paragraph mentioning the future areas of research.

Author Response

RESPONSES TO REFEREE 2 

The authors wish to express their thanks for the expert review of their manuscript. We have endeavored to address all of the points raised by the referee following the same order of the report. To ease the work of the referee, all the modifications are highlighted in yellow in the resubmitted version of the manuscript.

1. The authors have written Review Article well but it is too long and should have been 10-15 pages (excluding references).

This review article has 22 pages (excluding the abstract and references) and we agree that it is an extensive review. Nevertheless, since we attempted to address a broad spectrum of the cellular and soluble mediators involved in this disease, it is extremely difficult to reduce it. Therefore, in the “Concluding remarks” section, the main collected information has been summed up and gathered to ease the work to the potential readers.

2. Line #167. TLRs have also been associated with the development of NAFLD..... please check original Ref #35, it is mentioned NASH and not NAFLD. The authors should check and correct it.

We apologize for this mistake. The correction has been made (line 173).

3. Line#659. The Author should check Reference #104 where in original reference mentioned NASH and not NAFLD.

We appreciated your suggestion; although, according to this reference, NAFLD is mentioned and not NASH, as shown below:

“This study indicates that it is the accumulation of fat – both in the adipose tissue and in liver steatosis – that leads to increased hs CRP levels among obese patients. Thus, hs-CRP may be a marker of steatosis, but not of severity of NAFLD, in obese patients.”

4. Line#1014 Author should check the word NALF (typo error), it should be NAFL to NASH.

We apologize for this mistake. The correction has been made (line 1006).

5. Concluding remarks should have been in brief.

Thank you for your remark. We have shortened it a bit, following the reviewer’s recommendations. Additionally, we have placed the final conclusions after the new table 2, in order to make it easier for the readers.

6. The authors should have added a paragraph mentioning the future areas of research.

We really appreciated this comment. Therefore, we have included a table (Table 2) in which we have gathered some important future research topics in this area (page 21).
